# Structural Characterization and Effects on Insulin Resistance of a Novel Chondroitin Sulfate from *Halaelurus burgeri* Skin

**DOI:** 10.3390/md21040221

**Published:** 2023-03-30

**Authors:** Shiwei Hu, Hongli Zhu, Sichun Chen, Xiaofeng Wan, Yishu Liu, Zhaocai Ren, Shuang Gao

**Affiliations:** National Engineering Research Center for Maine Aquaculture, Zhejiang Ocean University, Zhoushan 316022, China

**Keywords:** chondroitin sulphate, *Halaelurus burgeri* skin, chemical structure, insulin resistance

## Abstract

Several studies have isolated chondroitin sulphate (CHS) from sharks’ jaws or cartilage. However, there has been little research on CHS from shark skin. In the present study, we extracted a novel CHS from *Halaelurus burgeri* skin, which has a novel chemical structure and bioactivity on improvement in insulin resistance. Results using Fourier transform–infrared spectroscopy (FT-IR), ^1^H-nuclear magnetic resonance spectroscopy (^1^H-NMR), and methylation analysis showed that the structure of the CHS was [4)-β-_D_-GlcpA-(1→3)-β-_D_-GlcpNAc-(1→]*_n_* with 17.40% of sulfate group concentration. Its molecular weight was 238.35 kDa, and the yield was 17.81%. Experiments on animals showed that this CHS could dramatically decrease body weight, reduce blood glucose and insulin levels, lower lipid concentrations both in the serum and the liver, improve glucose tolerance and insulin sensitivity, and regulate serum-inflammatory factors. These results demonstrated that the CHS from *H. burgeri* skin has a positive effect in reducing insulin resistance because of its novel structure, which provides a significant implication for the polysaccharide as a functional food.

## 1. Introduction

Insulin resistance occurs when normal circulating concentrations of insulin fail to regulate glucose uptake and utilization in cells because of the relative insufficiency of insulin secretion and the decreased sensitivity of target organs to the hormone. Insulin resistance eventually leads to the compensatory secretion of excessive insulin by the body, manifesting hyperinsulinemia to maintain body glucose homeostasis. Insulin resistance is closely related to the occurrence and development of obesity, hyperglycemia, type II diabetes, cardiovascular diseases, and other metabolic diseases [1]. Clinical studies have found that more than 80% of people with obesity show obvious insulin resistance, and the number of obese people in the world has exceeded 1 billion [2]. Among the 150 million patients with type II diabetes worldwide, the incidence of insulin resistance is more than 85% [3]. The prevalence of insulin resistance in developed countries is as high as 15–20%. It is predicted that insulin resistance will become a major cause of death due to chronic diseases around the world by 2050 [4]. Therefore, how to alleviate insulin resistance has become an important research topic.

Chondroitin sulfate (CHS) is a sulfated glycosaminoglycan formed by glucuronic acid (GlcA) and N-acetyl galactosamine (GalNAc) linked by alternating β1–3 or β1–4 glycosidic bonds [5]. Compared with CHS from terrestrial organisms, CHS from marine organisms mainly shows unique over-sulfuration characteristics, and the molecular structure type is regulated by the position of the sulfate group, such as C-4 (CHS-A), C-6 (CHS-C, more common), both C-4 and C-6 (CHS-E), C-6 of GalNAc and C-2 of GlcA (CHS-D), C-4 of GalNAc and C-2 of GlcA (CHS-B) [6]. Fish bone is an important source of marine CHS, and several papers have proved that CHS can be isolated from the jaw or cartilage [7,8]. Fish skin also contains abundant CHS, but there has been little research focusing on this topic. Only Nandini et al. reported that the CHS from *Prionace glauca* skin was highly heterogeneous and characterized by multiple disaccharides including GlcA-GalNAc, GlcA-GalNAc(6S), GlcA-GalNAc(4S), GlcA-GalNAc(4S,6S), GlcA(2S)-GalNAc(6S), and novel GlcA(2S)-GalNAc(4S), where 2*S*, 4*S*, and 6*S* represent 2-*O*-, 4-*O*-, and 6-*O*-sulfate, respectively [9]. A lack of investigation into CHS from shark skin has restricted its development and utilization. Moreover, the uses of marine CHS, including anti-arthritis [10], anti-neoplastic [11], anti-thrombotic and anti-coagulant [12], anti-diabetes [13], anti-hyperlipidemia [14], and neuroprotective [15] functions, have also attracted considerable attention in recent years. However, the effect of CHS from shark skin on insulin resistance has not been reported. Thus, in this study, the physicochemical properties and structural characteristics of polysaccharides from *Halaelurus burgeri* skin were systematically elucidated using high-performance liquid chromatography (HPLC), gas chromatography–mass spectrometer (GC-MS), Fourier transform–infrared spectroscopy (FT-IR), and nuclear magnetic resonance spectroscopy (NMR) ^1^D and ^2^D techniques. In addition, its effect was also investigated in in vivo phenotypes. This study recommends the use of shark skin polysaccharides as a food ingredient and pharmaceutical additive, providing information on novel therapies for insulin resistance and avenues for developing and applying pharmacological and food-based functional foods.

## 2. Results and Discussion

### 2.1. Isolation and Purification of CHS and General Property Description

The yield of crude polysaccharides isolated from dried *H. burgeri* skin was approximately 21.83%. First, a DEAE-52 anion exchange column was used to perform gradient elution (enrichment of neutral and acidic sugars). Four symmetrical peak-forming fractions were then collected using the phenol–sulfuric acid method, and the signal at A_490_ nm was detected by a microplate reader, yielding the following results: W: 2.2% (0.50 g), Y1: 5.1% (1.13 g), Y2: 3.8% (0.85), Y3: 21.8% (4.81 g) (Figure 1a). The outcomes demonstrated that Y3 had the highest yield and was shown to be a polysaccharide using the phenol–sulfuric acid method. The Y3 fraction was therefore used for further purification by a fully automated gel purification system (Figure 1b), where the symmetrical parts of the main peak were collected (140–160 min), with a final yield of 17.81%. The M_w_ of CHS determined by HPLC was 238.35 kDa, and the M_n_ was 143.72 kDa (Figure 1c, Table 1). Monosaccharide composition analysis showed that the CHS was composed of GalN; GlcN; Gal; Glc; GlcA (37.60; 3.92; 4.17; 6.95; 47.36) (Figure 1d, Table 1). The sulfate group content was determined to be 17.40% *via* ion chromatography (Figure 1e, Table 1). It was reported that the average molecular weight of the CHS from Tunisian fish skin was 41.72 kDa [16]. The CHS from cartilages of *Acipenser schrenckii* contained GlcN; GlcUA; GalN; Gal (1.4; 3.4; 3.7; 1.0) with a molecular weight of 299 kDa [17].

### 2.2. FT-IR Spectroscopy Analysis

We used FT-IR to examine the chemical functional groups of the CHS, and Figure 1f displays the information on the absorption peaks. The stretching vibration of -OH has a prominent peak at 3426.7 cm^−1^ [18]. The C-H stretching vibration is responsible for the little absorption peak at 2923.56 cm^−1^ [8]. The presence of acetylation is indicated by the absorption peak C=O carbonyl at 1637.2 cm^−1^ [17]. Uronic acid has been successfully decreased in the CHS without interfering with the subsequent methylation study, as indicated by the absence of an absorption peak at 1700 cm^−1^~1750 cm^−1^ [18]. The C-O stretching vibration is responsible for the absorption peaks at 1419.3 cm^−1^, 1230.36 cm^−1^, and 1128.15 cm^−1^, respectively [5]. The C=O molecule is responsible for the absorption peak at 1376.94 cm^−1^ [19]. The vibration of the sulfate group is given as the absorption peak at 1128.15 cm^−1^ [16]. The asymmetric pyran ring is responsible for the absorption peak at 929.99 cm^−1^ [19]. The presence of chondroitin sulfate-4 and the β-configuration is shown by the absorption peaks at 885.12 cm^−1^ and 855.97 cm^−1^, respectively.

### 2.3. Methylation and NMR Spectroscopy Analysis

The connection locations between monosaccharide residues were discovered by GC-MS to identify the types of glycosidic linkages, compared with standard mass spectrometric libraries (Figure 2a). Methylation analysis is more qualitative than quantitative because it is difficult to obtain standards for each individual monosaccharide derivative [12]. As shown in Table 2, there were two components of PMMA in the CHS: 2,3,4,6-Me4-Glcp and 2,3,6-Me3-Glcp. GlcpA residues are linked in a [→4)-β-_D_-GlcpA-(1→] aspect, while GlcpNAc residues are linked in a [→3)-β-_D_-GalpNAc-(1→] aspect.

The NMR method was used to analyze the chemical structure of the CHS. The proton signal of the sugar ring is present in the ^1^H NMR spectrum (Figure 2b) between δ3.2 and δ4.0 ppm, the chemical shift value of D_2_O is at δ4.70 ppm, and the signal peaks of the main terminal–matrix subpeaks δ4.41, δ4.47 are concentrated between 4.3 and 5.5 ppm. The chemical shift of the isomeric hydrogen at more than 5.0 ppm is considered as an alpha configuration, while at less than 5.0 ppm, it is considered as a β- configuration [20]. As shown in Figure 2b, the chemical shifts of the CHS were all less than 5.0 ppm, so the glycosidic bond type was β-configuration. The anomeric proton H−1 was in 4.5–5.5 ppm (4.61 ppm and 4.73 ppm), indicating that the CHS contained at least two kinds of sugar residues [21]. The signal of ^13^C spectrum is mainly concentrated between 60 and 120 ppm. We guessed that the CHS was in the β-configuration because the main anomeric carbon signal peaks in the measured ^13^C carbon spectrum were δ105.48, δ102.57 (Figure 2c), which were mainly dispersed between δ93 and δ105 ppm. This conclusion was proved in the results of the ^1^H NMR spectrum. Moreover, the carboxyl groups of uronic acid should be attributed to the signal peaks at δ176.34 and δ175.7 [22]. The anomeric carbon signal was δ105.47, and the corresponding anomeric proton was δ4.41, as shown in the HSQC spectra (Figure 2f). In the HH-COSY spectra (Figure 2d), the signals of H1-2, H2-3 and H3-4 were 4.41/3.29, 3.29/3.51, and 3.51/3.65, respectively. In conjunction with the NOESY spectra (Figure 2e), the findings demonstrated that δ4.41 was associated with δ3.61. H1/C1 was responsible for 4.41/105.47 ppm, followed by H2/C2 with 3.29/73.69 ppm, H3/C3 with 3.65/82.39 ppm, H4/C4 with 3.61/77.69 ppm, and C6 with a chemical shift of δ175 degrees. Hence, the signal was a part of the glycosidic bond→4)-β-_D_-GlcpA-(1→. It could be deduced from the results of the HH-COSY and NOESY spectra that 4.47/102.57 ppm belonged to H1/C1, 3.93/52.3 ppm to H2/C2, and 3.77/81.21 ppm to H3/C3. The signal should therefore be associated with the glycosidic bond →3)-β-_D_-GlcpNAc-(1→. Similarly, all the glycosidic– bond signals were categorized in accordance with the law by NOESY (Table 3).

According to the one-dimensional and two-dimensional NMR spectra, we assigned the glycosidic–bond signals of the polysaccharides as →4)-β-_D_-GlcpA-(1→ and H3 of →3)-β-_D_-GlcpNAc-(1→. This showed that there was a glycosidic bond →4)-β-_D_-GlcpA-(1→3)-β-_D_-GlcpNAc-(1→. The anomeric proton of the glycosidic bond →3)-β-_D_-GlcpNAc-(1→ had a signal peak with H4 of →4)-β-_D_-GlcpA-(1→, showing that there were the glycosidic bonds →3)-β-_D_-GlcpNAc-(1→4)-β-_D_-GlcpA-(1→.

Most focus signals could not be recognized because the signal was covered by other sugars due to the low content of focus. In the GC-MS methylation analysis, the specific relationship between peak and content could not be determined because of the different responsivity of each glycosidic bond without a methylation standard [23]. Therefore, methylation analysis is generally not used as a quantitative standard. We obtained the structural formula of the CHS as [4)-β-_D_-GlcpA-(1→3)-β-_D_-GlcpNAc-(1→]_n_ through complete analysis with monosaccharide components, the type of methylated–glycosidic bonds, ion chromatography, and the NMR spectrum (Figure 3). Its structure is similar to the CHS from shark jaw or cartilage [4,5].

### 2.4. CHS Reduced Body Weight

As shown in Table 4, after 18 weeks of feeding with CHS, body weight gain, the ratio of liver to body weight, and kidney to body weight were all significantly reduced in the CHS-treated mice compared with the HFD animals (*p* < 0.05). The rate of epididymal fat to body weight was also remarkably decreased by 83.39% (*p* < 0.05). Meanwhile, H&E strain results showed that HFD-induced epididymal adipocyte infiltration of inflammatory cells, proliferation, and deformation were all improved in the CHS group (Figure 4A–D).

### 2.5. CHS Decreased Blood Glucose, Insulin, and Improvement of Glucose Tolerance and Insulin Tolerance

As shown in Table 4, the CHS caused dramatic decreases in fasting blood glucose and serum insulin levels by 60.42% and 13.63% compared with the HFD mice (*p* < 0.05). CHS treatment showed a 37.73% decrease in the HOMA-IR score and a 60.53% increase in the QUICKI value, respectively. Moreover, the CHS markedly improved impaired glucose tolerance through the experiment of OGTT (*p* < 0.05, Figure 4a,b), and the sensibility of insulin to deal with glucose with the reduction of AUC_IITT_ (*p* < 0.05, Figure 4c,d). These indicate that CHS can increase insulin sensibility and alleviate insulin resistance. 

### 2.6. CHS Reduces the Abnormal Lipids in Serum and in the Liver

We detected the actuation atherosclerotic lipid index (TG, TC, LDL-C) and anti-atherosclerosis lipid index (HDL-C) in the blood lipids of the HFD mice to examine recovery from dyslipidemia after the CHS treatment. As depicted in Table 4, HDL-C was increased by 72.61% while serums TC, TG, and LDL-C were reduced by 32.84%, 46.07%, and 23.76%, respectively (*p* < 0.05) in the CHS-treated mice compared with the HFD animals. These findings demonstrate that CHS can reduce dyslipidemia in the HFD mice and maintain lipid metabolism and cholesterol levels in the body.

Excessive liver fat content is the major characteristic of fatty liver disease linked to obesity and diabetes [24]. We evaluated hepatic TC, TG, glutamic oxalacetic transaminase (AST), and glutamic pyruvic transaminase (ALT) levels in the HFD mice treated with CHS (Table 4). Oil red O and H&E staining was also tested to indicate the accumulation of a high number of triglyceride lipid droplets and inflammatory cell infiltration lesions. The HFD mice given CHS showed significant reductions in TC, TG, and AST/ALT levels by 48.53%, 34.18%, and 18.27% (*p* < 0.05), respectively, in comparison to those in insulin-resistant mice. As shown in Figure 4E–H, the buildup of orange lipid droplets in liver cells was significantly decreased in the CHS-treated mice compared with the HFD group. However, the amount of lipid droplets in the melbine (DMBG) group was substantial, suggesting that the effect of DMBG lipolysis on the liver was less than that of CHS. These indicate that CHS can alleviate the enzymological anomalies of a metabolic fatty liver caused by insulin resistance.

### 2.7. CHS Attenuated Circulatory Inflammatory Cytokines

We measured four insulin-resistance-related inflammatory cytokines, including pro-inflammatory factors (interleukin−1β (IL−1β), IL-6, and tumor necrosis factor-a (TNF-a)) and anti-inflammatory factor (IL−10). The pro-inflammatory factors in the HFD group were significantly greater than those in the normal group, whereas the anti-inflammatory factor level was the opposite, showing aberrant serum inflammatory cytokines levels in the insulin-resistant animals. When treated with CHS, insulin-resistant mice showed remarkable reductions in serum IL−1β, IL-6, and TNF-a concentrations, and a significant elevation in serum IL−10 concentration (*p* < 0.05). Increased adipocyte directly secretes pro-inflammatory cytokines in insulin-resistant individuals [25]. Moreover, high levels of IL−1β, IL-6, and TNF-a concentrations can inhibit insulin receptors and their substrates activities through increasing serine phosphorylation, while IL−10 can promote these activities [26]. These indicate that CHS can regulate inflammatory cytokines in insulin-resistant mice.

Several papers have proved that the CHS from marine species could improve insulin resistance or alleviate inflammation. For example, Chen et al. reported that CHS could improve insulin sensitivity through activating AKT signaling [27]. Fucoidan from brown seaweed could suppress inflammation by blocking the NF-κB signal pathway [28]. Our previous studies also proved that the CHS from sea cucumbers could improve insulin resistance by activating PI3K/AKT signaling [29,30]. In the present study, the CHS from shark skin also mitigated insulin resistance and circulatory inflammatory cytokines. Further studies on the mechanism will be investigated.

## 3. Materials and Methods

### 3.1. Extraction and Purification of the CHS

The extraction of the CHS was based on the previous method with some modifications [29]. Dried sharkskin powder was first defatted by soaking in acetone (v/m, 1:10) to produce crude sulfated polysaccharides. The precipitate was then digested by papain-based enzymatic digestion, precipitation with 10% CPC, and dialysis using a 10,000 Da dialysis bag (named CHS). The CHS was further separated and eluted in a gradient of three column volumes of water (designated as W), 0.2 M NaCl (Y1), 0.5 M NaCl (Y2), and 1.0 M NaCl (Y3). The Y3 was detected by the phenol–sulfuric acid method to confirm it as CHS and was purified in an automatic gel purification system (Yangzhou, China), concentrated, and lyophilized to obtain pure CHS. The yield was calculated.

### 3.2. Determination of General Properties

HPLC was used to detect monosaccharide components (Shimadzu, Tokyo, Japan) with a Dionex Carbopac TMPA20 column (3 × 150 mm, ThermoFisher, Shanghai, China). The molecular weight of the CHS was determined by gel permeation chromatography. Ion chromatography (ThermoFisher, ICS 5000, Shanghai, China) was used to measure the sulphate content using anhydrous sodium sulphate as the standard.

### 3.3. FT-IR Spectroscopy Analysis

The components were mixed with KBr, crushed, and compressed into powder using a Fourier transform–infrared spectrometer (FT-IR650, Gangdong, Tianjin, China). The CHS spectra were then captured and scanned in the 4000–400 cm^−1^ wavenumber region [31].

### 3.4. Methylation and NMR Analysis

Methylation analysis was carried out to determine the types of glycosidic bonds present by GC-MS with reference to the method after reduction with aldehyde acid [32]. Briefly, following methylation, TFA hydrolysis, reduction, and acetic anhydride–pyridine acetylation, the CHS samples were subjected to GC-MS (Shimadzu, GCMS-QP 2010, Tokyo, Japan) for analysis.

NMR analysis was used to analyze the chemical structure of the CHS. Briefly, 3 mg of the CHS was completely dissolved in 1 mL 99.9% D_2_O at room temperature for 12 h. NMR was used to examine ^1^D NMR (^1^H NMR spectra, 13C NMR spectra), ^2^D NMR (COSY spectra, HSQC spectra, and NOESY spectra) on a 600 MHz and 10 °C at 1 g/L in D_2_O.

### 3.5. Animal Experiments

C57BL/6J mice (SPF, male, 8 weeks, 18 ± 2 g) were provided by Spelford Laboratory Animal Technologies Ltd. (SCXK2019-0001, Beijing, China). The mice were housed in an equal balance of 12/12 h of light/dark at 23 ± 1 °C daily and in individual cages. All procedures in this study were reviewed and approved by the Ethics Committee at Zhejiang Ocean University (No. 20210310). The study’s staff members undertook thorough training and received the required permissions before conducting the animal tests.

The animals were divided into four groups (*n* = 10): A control group (fed with a standard diet), an HFD group (fed with a high-fat diet (HFD) of 29% carbohydrates, 16% protein, and 55% fat), a CHS group (maintained with HFD and 80 mg/kg of CHS intragastrically), and a melbine (DMBG) group (administrated with HFD and 40 mg/kg of DMBG intra-gastrically). After 17 weeks of continuous treatment, blood glucose and insulin were measured to decide whether the insulin-resistant model was successful. Meanwhile, OGTT and IITT experiments were performed with 5 animals per experiment. The mice were sacrificed at week 18 to obtain blood for subsequent analysis.

### 3.6. Insulin-Resistance-Related Parameters in Blood and Liver Measurement

Fasting blood glucose, serum TC, TG, HDL-C, and LDL-C, and hepatic TC and TG levels were measured using commercial kits (Biosino, Beijing, China). Serum insulin, IL−1β, IL-6, TNF-a, and IL−10 concentrations and hepatic AST and ALT concentrations were tested using ELISA kits (Invitrogen, Carlsbad, CA, USA). HOMA-IR and QUICKI values were calculated using Equations (1) and (2), respectively.
HOMA-IR = (fasting blood glucose × serum insulin)/22.5(1)
QUICKI = 1/[lg(fasting blood glucose) + lg(serum insulin)](2)

### 3.7. OGTT and IITT

After 17 weeks of feeding, the OGTT experiment was performed by measuring the blood glucose levels of 6 h fasted mice at 0, 0.5, 1, and 2 h after intra-gastric administration of 2 g/kg glucose (*n* = 5). Residual animals were used to carry out the IITT experiment similarly to the OGTT with an intra-peritoneal injection of 0.5 U/kg insulin. Blood glucose levels were measured using a commercial kit. The areas under the curve of the OGTT and IITT (AUC_OGTT/IITT_) were both calculated using Equation (3).
AUC_OGTT_/AUC_IITT_ = 0.25 × A + 0.5 × B +0.75 × C + 0.5 × D (3)

A, B, C, and D represent blood glucose levels at 0, 0.5, 1, and 2 h after treating with glucose or insulin, respectively).

### 3.8. H&E and Oil Red O Strain

Epididymal fat and liver tissues were fixed in 4% formaldehyde, embedded in paraffin, and subsequently cut into 5 µm thick sections. After deparaffinization and hydration, the sections of epididymal fat tissues were stained with H&E to assess fat morphology. Oil red O staining was used to demonstrate lipid droplets in the liver. Microscopic structures of the epididymal adipose and the liver were observed and photographed using a fluorescence microscope (Eclipse Ci, Nikon, Japan).

### 3.9. Statistical Analysis

Data are expressed as the mean ± standard deviation (SD). One-way analysis of variance (ANOVA) followed by Student’s *t*-test were performed for significant difference among the four groups with SPSS 21.0 software. Statistical significance was considered at *p* < 0.05.

## 4. Conclusions

We herein demonstrated that the chemistry structure of a novel CHS from *H. burgeri* skin was described as [4)-β-_D_-GlcpA-(1→3)-β-_D_-GlcpNAc-(1→]*_n_* with 17.40% of sulfate. Its molecular weight was 238.35 kDa, and the yield was 17.81%. Insulin resistance was improved in the HFD mice when treated with the CHS, with reductions in body weight, blood glucose, lipids in serum and the liver, and regulation of inflammatory cytolines.

## Figures and Tables

**Figure 1 marinedrugs-21-00221-f001:**
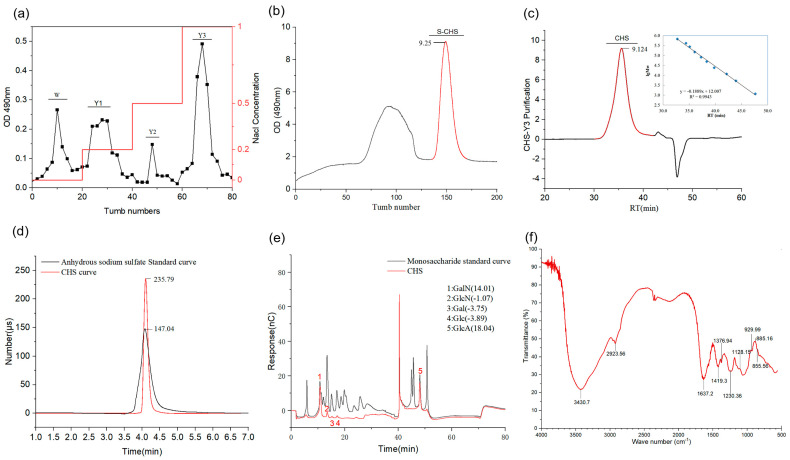
Isolation, purification, and general property analysis of the CHS. (**a**) Stepwise–elution curve of the CHS on a DEAE-52 ion exchange column. (**b**) Purification of the CHS by Automatic Gel Purification System. (**c**) HPGPC of the CHS (molecular weight). (**d**) HPLC diagram of the monosaccharide composition of the CHS. (**e**) Sulfate group content by ion chromatography. (**f**) FT-IR analysis.

**Figure 2 marinedrugs-21-00221-f002:**
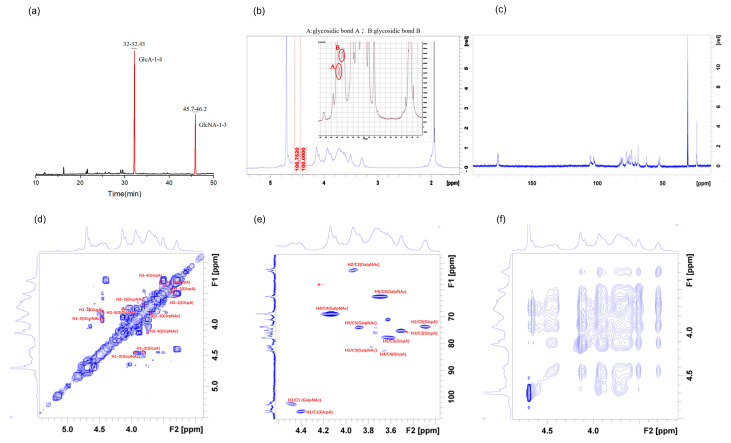
Structural characterization of the CHS: (**a**) polysaccharide methylation; (**b**) ^1^H spectrum; (**c**) ^13^C spectrum; (**d**) HH-COSY spectrum; (**e**) HSQC spectrum; (**f**) NOESY spectrum.

**Figure 3 marinedrugs-21-00221-f003:**
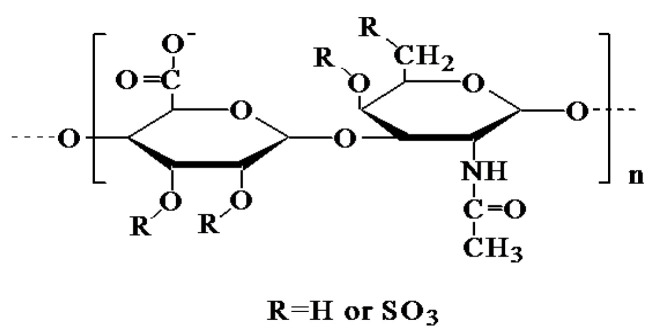
Chemical structural of the CHS.

**Figure 4 marinedrugs-21-00221-f004:**
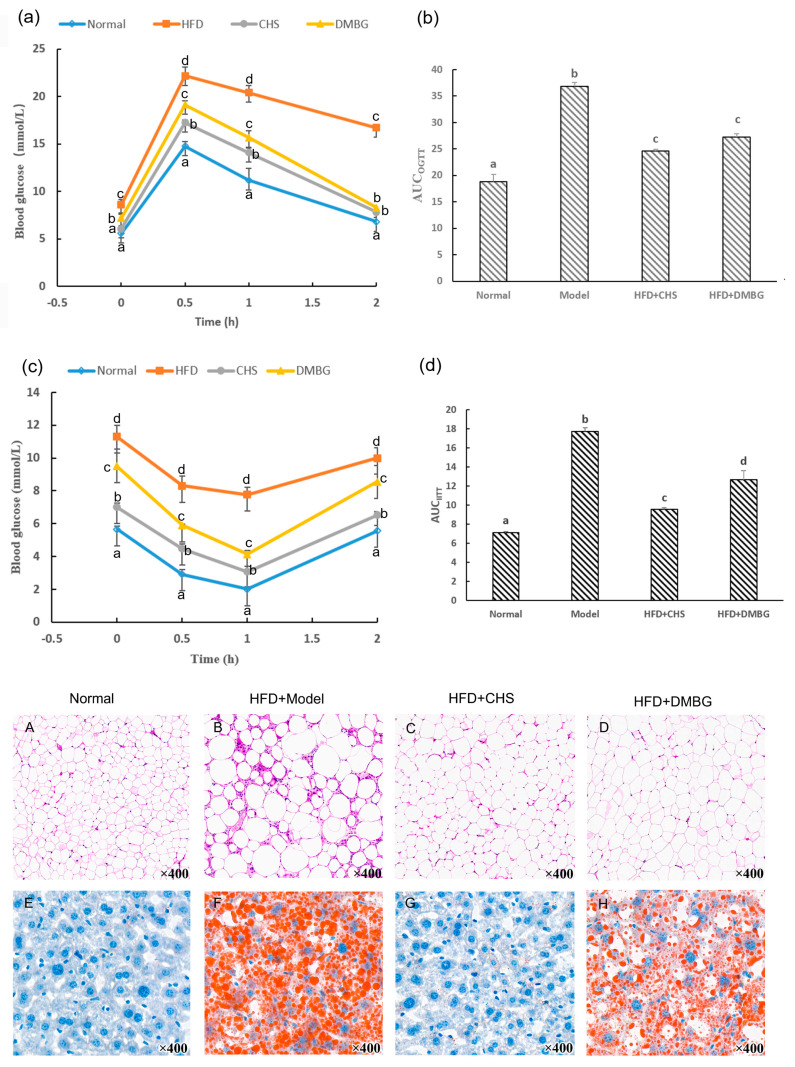
Effects of the CHS on OGTT and IITT, epididymal fat H&E strain, and liver oil red O strain in insulin-resistant mice (*n* = 10). (**a**) Blood glucose levels in OGTT. (**b**) AUC_OGTT_. (**c**) Blood glucose levels in IITT. (**d**) AUC_IITT_. (**A**–**D**) H&E staining for the epididymal fat tissues (×400). (**E**–**H**) Oil red O staining for the liver tissues (×400). Different lowercases represented significant difference (*p* < 0.05) compared between groups. CHS: chondroitin sulfate; DMBG: melbine; H&E: hematoxylin and eosin; IITT: intra-peritoneal insulin tolerance test; ORO: oil red O; OTGG: oral glucose tolerance test.

**Table 1 marinedrugs-21-00221-t001:** Component characterization of CHS.

Sample	Monosaccharide Composition Proportion %					Sulfate	M_p_(kDa)	M_w_(kDa)	M_n_(kDa)	Polydispersity
	GalN	GlcN	Gal	Gac	GlcA					
CHS	37.60	3.92	4.17	6.95	47.36	17.40%	173.75	238.35	143.72	1.66

**Table 2 marinedrugs-21-00221-t002:** Analysis of the CHS methylation.

RT	Methylated Sugar	Mass Fragments (*m*/*z*)	Area Ratios	Type of Linkage
32.180	2,3,4,6-Me4-Glcp	43,87,99,101,113,117,129,131,161,173,233	69.69	→4)-β-d-GlcpA-(1→
45.879	2,3,6-Me3-Galp	43,75,100,117,129,158,171	30.31	→3)-β-d-GalpNAc-(1→

**Table 3 marinedrugs-21-00221-t003:** The attribution of ^1^H and ^13^C signals.

	Glycosyl Residues	H1/C1	H2/C2	H3/C3	H4/C4	H5/C5	H6/C6	C-Me	C=O
Glycosidic bond A	→4)-β-_D_-GlcpA-(1→	4.41	3.29	3.51	3.65	3.61			
		105.47	73.69	75.11	82.39	77.69	175.69		
Glycosidic bond B	→3)-β-_D_-GlcpNAc-(1→	4.47	3.93	3.77	4.14	3.89	3.7		
		102.56	52.29	81.20	68.85	73.90	62.36	23.90	176.33

**Table 4 marinedrugs-21-00221-t004:** Effects of the CHS on insulin-resistance-related parameters in the HFD mice.

	Normal	HFD	HFD + CHS	HFD + DMBG
Body weight gain (g)	10.93 ± 0.97 ^a^	20.62 ± 1.23 ^a^	9.85 ± 0.98 ^c^	11.81 ± 0.65 ^d^
Liver/body weight (g/100g)	3.36 ± 0.25 ^a^	7.32 ± 0.84 ^b^	4.29 ± 0.41 ^c^	4.73 ± 0.56 ^c^
Kidney/body weight (g/100g)	1.06 ± 0.09 ^ab^	1.82 ± 0.12 ^b^	1.42 ± 0.16 ^c^	1.28 ± 0.12 ^c^
Abdominal fat/body weight (g/g)	0.47 ± 0.123 ^c^	2.426 ± 0.43 ^a^	0.403 ± 0.106 ^b^	1.376 ± 0.798 ^d^
Fasting blood glucose (mmol/L)	5.21 ± 0.52 ^a^	13.44 ± 0.32 ^b^	5.32 ± 0.84 ^a^	6.07 ± 0.84 ^a^
Serum insulin (mIU/L)	25.99 ± 0.1 ^a^	32.64 ± 1.1 ^c^	28.19 ± 0.36 ^b^	28.47 ± 1.03 ^b^
HOMA-IR	6.36 ± 0.39 ^a^	12.22 ± 0.72 ^b^	7.61 ± 0.46 ^c^	8.77 ± 0.87 ^d^
QUICKI	0.007 ± 0.0004 ^a^	0.004 ± 0.0002 ^b^	0.006 ± 0.0004 ^c^	0.005 ± 0.0005 ^c^
Serum TC (mmol/L)	3.75 ± 0.76 ^a^	7.35 ± 0.30 ^c^	4.90 ± 0.16 ^b^	4.89 ± 0.26 ^ab^
Serum TG (mmol/L)	1.37 ± 0.009 ^a^	2.66 ± 0.054 ^c^	1.43 ± 0.152 ^ab^	1.62 ± 0.061 ^b^
Hepatic TC (mmol/L)	0.005 ± 0.001 ^a^	0.009 ± 0.001 ^c^	0.005 ± 0.001 ^a^	0.006 ± 0.001 ^b^
Hepatic TG (mmol/L)	0.018 ± 0.005 ^a^	0.026 ± 0.001 ^b^	0.017 ± 0.004 ^a^	0.020 ± 0.003 ^ab^
Serum HDL-C (mmol/L)	1.30 ± 0.09 ^a^	0.31 ± 0.023 ^b^	1.17 ± 0.08 ^c^	0.44 ± 0.09 ^d^
Serum LDL-C (mmol/L)	2.50 ± 0.20 ^a^	5.26 ± 0.37 ^b^	4.01 ± 0.4 ^c^	4.64 ± 0.28 ^d^
Serum IL-6 (pg/mL)	55.93 ± 1.05 ^a^	78.98 ± 1.51 ^c^	64.48 ± 1.16 ^b^	57.50 ± 0.83 ^a^
Serum IL−1β (pg/mL)	49.28 ± 1.53 ^a^	81.17 ± 1.79 ^b^	73.84 ± 1.53 ^c^	70.95 ± 2.01 ^d^
Serum IL−10 (pg/mL)	274.06 ± 1.27 ^a^	256.05 ± 2.36 ^b^	272.74 ± 2.37 ^a^	264.82 ± 1.61 ^c^
Serum TNF-α (pg/mL)	19.52 ± 3.29 ^a^	106.01 ± 3.83 ^b^	63.83 ± 5.19 ^c^	35.9 ± 2.39 ^d^
Hepatic ALT (U/L)	50.19 ± 3.65 ^a^	39.82 ± 1.66 ^b^	47.41 ± 0.79 ^a^	40.59 ± 2.84 ^b^
Hepatic AST (U/L)	159.90 ± 9.79 ^a^	114.49 ± 14.83 ^c^	136.19 ± 13.17 ^b^	146.80 ± 8.10 ^ab^

Note: Data are presented as mean ± S.D. (*n* = 10). Multiple comparisons were made using one way ANOVA. Different lowercases represent significant difference (*p* < 0.05) compared between groups. ALT: glutamic pyruvic transaminase; AST: glutamic oxalacetic transaminase; CHS: chondroitin sulfate; DMBG: melbine; HDL-C: high-density lipoprotein cholesterol; HFD: high fat diet; HOMA-IR: homeostasis model assessment of insulin resistance; IL: interleukin; LDL-C: low-density lipoprotein cholesterol; QUICKI: quantitative insulin sensitivity index; SD: standard deviation; TC: total cholesterol; TG: triglyceride; TNF-a: tumor necrosis factor-a.

## Data Availability

The original data presented in the study are included in the article; further inquiries can be directed to the corresponding author.

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
