# Peer review of "Structural Characterization and Effects on Insulin Resistance of a Novel Chondroitin Sulfate from Halaelurus burgeri Skin"

_marinedrugs, 2023, doi:10.3390/md21040221_

Round 1

Reviewer 1 Report

Comments to the Author

The authors have researched on the structure of chondroitin sulfate derived from fish skin and its activity in improving insulin resistance. However, the paper is not prepared carefully. There are too many mistakes, and specific comments as follows:

1. Compared with chondroitin sulfate extracted from cartilage, what are the properties of chondroitin sulfate derived from fish skin? Why study chondroitin sulfate from fish skin?

2. 2.1: Y3 fraction of polysaccharides was detected by phenol-sulfuric acid method. Please describe this in the “materials and method”.

3. What is the highest peak in Figure 1(d). The peaks of the monosaccharide standards should also be marked. Table 1: It is proportion instead of ratio? How was the monosaccharide composition analyzed? Please describe this in the “materials and method”. In addition, GlcA seems too much. By the way, GlcUA(Line 86) or GlcA (Table 1)?

4. Figure 1: (a)  Please change “NACLto “NaCl”. Please mark the absorption peaks at 885cm-1 and 856 cm-1 in the Fig.1f.

5. 2.4: Why did the liver and kidney weight increase significantly in the HFD+CHS and HFD+DMBG treatment groups, compared to the normal and HFD groups?

6. There are no significant difference symbols marked in Figure 3a and c. The significance labels (a, b, c, d) are not marked in a correct way.

7. Line 193-194: What is missing before %?

8. Please provide the full name of DMBG.

9. Table 4. Please indicate the meaning of the uppercase letters (a,b,c…)

10. Table 2. What the methylated sugar instead of GlcpA? Is it reduced? In addition, the ratio of GlcpA to GalpNAc is 2:1 instead of 1:1. This is not consistent with the structure of CHS.

11. Line 117: a reference citation is missing.

12. 3.5: For the experimental groups, how to prove the insulin resistant model induced by HFD was successful?

Author Response

Friday, 24 March 2023

Marine Drugs

Dear reviewer,

Thank you for your effort on our manuscript, namely “Structural characterization and effects on insulin resistance of a novel chondroitin sulfate from Halaelurus burgeri skin”. We also appreciate the meticulous review provided by your expert. Our responses to the comments are as follows.

Reviewer 1

The authors have researched on the structure of chondroitin sulfate derived from fish skin and its activity in improving insulin resistance. However, the paper is not prepared carefully. There are too many mistakes, and specific comments as follows:

  1. Compared with chondroitin sulfate extracted from cartilage, what are the properties of chondroitin sulfate derived from fish skin? Why study chondroitin sulfate from fish skin?

 Answer: Thank you very much for the suggestions. The structure of CHS from shark skin is similar to that from shark jaw or cartilage, and we have explained it in line 147. Fish skin amount is abundant and contains abundant CHS, but little researches focused on it. Short of investigations on CHS from shark skin restrict its development and utilization. Therefore, we studied CHS from shark skin

  1. 2.1: Y3 fraction of polysaccharides was detected by phenol-sulfuric acid method. Please describe this in the “materials and method”.

 Answer: Thank you very much for the advice. We have added the description in line 230-232.

  1. What is the highest peak in Figure 1(d). The peaks of the monosaccharide standards should also be marked. Table 1: It is proportion instead of ratio? How was the monosaccharide composition analyzed? Please describe this in the “materials and method”. In addition, GlcA seems too much. By the way, GlcUA(Line 86) or GlcA (Table 1)?

 Answer: Thank you very much for the advice. We have check the data of monosaccharide compositions, and corrected it in Table 1. The monosaccharide composition was analyzed using HPLC, and we have explained it in line 215-216.

  1. Figure 1: (a) Please change “NACL” to “NaCl”. Please mark the absorption peaks at 885cm-1 and 856 cm-1 in the Fig.1f.

 Answer: Thank you very much for the suggestion. We have changed “NACL” to “NaCl” in Fig.1a, and marked the absorption peaks at 885cm-1 and 856 cm-1 in the Fig.1f.

  1. 2.4: Why did the liver and kidney weight increase significantly in the HFD+CHS and HFD+DMBG treatment groups, compared to the normal and HFD groups?

 Answer: Thank you very much for the suggestion. We have checked it and corrected the data of the liver and kidney weight in Table 4.

  1. There are no significant difference symbols marked in Figure 3a and c. The significance labels (a, b, c, d) are not marked in a correct way.

 Answer: Thanks for the suggestion. We have added significant difference symbols marked in Fig.3a and c.

  1. Line 193-194: What is missing before %?

 Answer: Thank you very much for the suggestions. We have added the data before %.

  1. Please provide the full name of DMBG.

 Answer: Thank you very much for the suggestions.

  1. Table 4. Please indicate the meaning of the uppercase letters (a,b,c…)

 Answer: Thank you very much for the suggestions. We have added the information after Note.

  1. Table 2. What the methylated sugar instead of GlcpA? Is it reduced? In addition, the ratio of GlcpA to GalpNAc is 2:1 instead of 1:1. This is not consistent with the structure of CHS.

 Answer: Thank you very much for the suggestions. In GC-MS methylation analysis, the specific relationship between peak and content could not be determined because of different responsivity of each glycosidic bond without methylation standard. Therefore, methylation analysis is generally not as a quantitative standard. And we cited a reference [23] to prove it. The explanation is shown in line 142-144.

  1. Line 117: a reference citation is missing.

 Answer: Thank you very much for the suggestions. We have added the reference.

  1. 3.5: For the experimental groups, how to prove the insulin resistant model induced by HFD was successful?

Answer: Thank you very much for the suggestions. The insulin resistance model have successful established in our previous studies. Significant increases in blood glucose, insulin, impaired glucose tolerance and insulin tolerance synthetically proved the model was successfully established at Week 17, and we have explained it in line 258-260.

Thank you for your consideration. I look forward to hearing from you.

Best wishes

Yours sincerely,

Shiwei Hu

Reviewer 2 Report

The manuscript shows the structural characterization of a chondroitin sulfate from shark skin and its pharmacological effects in an insulin resistance model in mice. The topic is important and of interest to Marine Drugs readers, however, a major revision must be done to be considered for publication.

General considerations: 

1-    English must be proofread for grammar and spelling. “vulcanized at different carbon positions” (line 41), “Protecting nerves activities” (line 55), “have significant significance.” (line 190) are some examples. What parameters do authors consider when they claim that “The molecular homogeneity was good.”?!? (line 79).

2-    Figure legends should contain more information about the experiment. In addition, the authors use many abbreviations throughout the text and may consider an abbreviation list to facilitate understanding

3-    Materials and Methods: 

3.1- The text requires an English revision and the addition of references in many sections. The methodology is confusing and contains numerous writing errors, such as Line 247 where "20 L of 0.05 M NaCl solution was injected into the column," which should likely be 20 uL. Additionally, there are experimental inconsistencies, such as the use of deuterated acetone as an internal reference and chemical shift correction in Line 265. Was D2O and deuterated acetone actually used in the NMR, and was there any conflict of deuterium when locking? Furthermore, technical data on the execution of experiments, such as the gradient used in Carbopac, temperature, detection method (optical or electrochemical), detection rate, and molecular weight standards in SEC, are missing. These omissions apply to all experiments. 

3.2- Authors must explain what would be the “melbine group (DMBG)”? Is there a specific fraction used in 2% CHS group? Was chondroitin added to the feed and was there any estimation of how much was consumed?

4-    Results: 

4.1- The text contains English errors and requires revision. Throughout the text, the author makes various claims, such as "The asymmetric ring tensile vibration of the pyran ring is responsible for the absorption peak at 929.99 cm–1" (lines 100 and 101) and "The examination of proportion in NMR analysis typically takes into account the hydrogen spectrum" (lines 100 and 101), without providing a reference on the topic. This issue is repeated throughout the text. The terms "Heterocarbon" and "Heterohydorgen" should be replaced with "anomeric carbon" and "anomeric protons," respectively. 

4.2- Line 159 and 160 state, "Similarly, all glycosidic bond signals are categorized in accordance with the law by combining HMBC and NOESY, as shown in Tab.3." Was HMBC actually performed, as there is no record in the methodology?    

 4.3- The correct term for NMR experiments is "spectra", not "charts" or "maps". 

4.4- Regarding the figures, there are some issues that need to be addressed. Firstly, the panels do not follow a consistent standard and there are missing axis labels for panels 1D and 1E. It is recommended to redo the figures following a consistent format for all panels.  In addition, panels B, C, and F are missing labels, making it difficult to understand the information presented. The legends for panels E and F are also incorrect and It is not clear what "A" and "B" highlighted in panel 2B refer to, which should be clarified. Overall, it is important to ensure that figures are clear, organized, and easy to understand for readers. 

4.5- It is important to have a Figure showing the chemical structure of chondroitin sulfate from shark skin for better visualization.

4.6- The authors can better discuss the pharmacological effect observed with other works in the literature using fucosylated chondroitin sulfate from marine invertebrates or even compare the anti-inflammatory activity of chondroitin sulfate from mammals. Is there any hypothesis about the mechanism of action? The discussion is very poor in this aspect.

1- Liu, H.H.; Ko,W.C.; Hu, M.L. Hypolipidemic Effect of Glycosaminoglycans from the Sea Cucumber Metriatyla Scabra in Rats Fed a Cholesterol-Supplemented Diet. J. Agric. Food Chem. 2002, 50, 3602–3606

2- Yin, J.; Wang, J.; Li, F.; Yang, Z.; Yang, X.; Sun, W.; Xia, B.; Li, T.; Song, W.; Guo, S. The Fucoidan from the Brown Seaweed: Ascophyllum Nodosum Ameliorates Atherosclerosis in Apolipoprotein E-Deficient Mice. Food Funct. 2019, 10, 5124–5139.

3- Hu, S.; Chang, Y.; He, M.; Wang, J.; Wang, Y.; Xue, C. Fucosylated Chondroitin Sulfate from Sea Cucumber Improves Insulin Sensitivity via Activation of PI3K/PKB Pathway. J. Food Sci. 2014, 79, H1424–H1429.

4- Hu, S.; Chang, Y.;Wang, J.; Xue, C.; Li, Z.;Wang, Y. Fucosylated Chondroitin Sulfate from Sea Cucumber in Combination with Rosiglitazone Improved Glucose Metabolism in the Liver of the Insulin-Resistant Mice. Biosci. Biotechnol. Biochem. 2013, 77,

2263–2268.

Author Response

Friday, 24 March 2023

Marine Drugs

Dear reviewer,

Thank you for your effort on our manuscript, namely “Structural characterization and effects on insulin resistance of a novel chondroitin sulfate from Halaelurus burgeri skin”. We also appreciate the meticulous review provided by your expert. Our responses to the comments are as follows.

Reviewer 2

The manuscript shows the structural characterization of a chondroitin sulfate from shark skin and its pharmacological effects in an insulin resistance model in mice. The topic is important and of interest to Marine Drugs readers, however, a major revision must be done to be considered for publication.

General considerations:

1-    English must be proofread for grammar and spelling. “vulcanized at different carbon positions” (line 41), “Protecting nerves activities” (line 55), “have significant significance.” (line 190) are some examples. What parameters do authors consider when they claim that “The molecular homogeneity was good.”?!? (line 79).

 Answer: Thank you very much for the suggestions. We have edited the English under the help of a professional editorial company.

2-    Figure legends should contain more information about the experiment. In addition, the authors use many abbreviations throughout the text and may consider an abbreviation list to facilitate understanding

 Answer: Thank you very much for the suggestions. We have added informations about the experiments in figure legends. And we have added an abbreviation list in line 304-311.

3-    Materials and Methods:

3.1- The text requires an English revision and the addition of references in many sections. The methodology is confusing and contains numerous writing errors, such as Line 247 where "20 L of 0.05 M NaCl solution was injected into the column," which should likely be 20 uL. Additionally, there are experimental inconsistencies, such as the use of deuterated acetone as an internal reference and chemical shift correction in Line 265. Was D2O and deuterated acetone actually used in the NMR, and was there any conflict of deuterium when locking? Furthermore, technical data on the execution of experiments, such as the gradient used in Carbopac, temperature, detection method (optical or electrochemical), detection rate, and molecular weight standards in SEC, are missing. These omissions apply to all experiments.

 Answer: Thanks for the advices. We have added the missing references. And we have edited an English revision. We have also added the information on the experiments.

3.2- Authors must explain what would be the “melbine group (DMBG)”? Is there a specific fraction used in 2% CHS group? Was chondroitin added to the feed and was there any estimation of how much was consumed?

 Answer: Thanks for the advices. We have corrected the dosage as 80 mg/kg of CHS intragastrically and 40 mg/kg of DMBG intragastrically. The dosage of CHS is according to our data of preliminary experiment, while the dosage of DMBG is according to recommended dose of the drug.

4-    Results:

4.1- The text contains English errors and requires revision. Throughout the text, the author makes various claims, such as "The asymmetric ring tensile vibration of the pyran ring is responsible for the absorption peak at 929.99 cm–1" (lines 100 and 101) and "The examination of proportion in NMR analysis typically takes into account the hydrogen spectrum" (lines 100 and 101), without providing a reference on the topic. This issue is repeated throughout the text. The terms "Heterocarbon" and "Heterohydorgen" should be replaced with "anomeric carbon" and "anomeric protons," respectively.

 Answer: Thank you very much for the suggestions. We have added the references. And we have corrected "Heterohydorgen" with "anomeric carbon" and "anomeric protons," respectively.

4.2- Line 159 and 160 state, "Similarly, all glycosidic bond signals are categorized in accordance with the law by combining HMBC and NOESY, as shown in Tab.3." Was HMBC actually performed, as there is no record in the methodology?   

 Answer: Thank you very much for the suggestions. We have corrected it as “all glycosidic bond signals were categorized in accordance with the law by NOESY (Table 3)” in line 130-131.

4.3- The correct term for NMR experiments is "spectra", not "charts" or "maps".

 Answer: Thank you very much for the suggestions. We have corrected it as "spectra".

4.4- Regarding the figures, there are some issues that need to be addressed. Firstly, the panels do not follow a consistent standard and there are missing axis labels for panels 1D and 1E. It is recommended to redo the figures following a consistent format for all panels.  In addition, panels B, C, and F are missing labels, making it difficult to understand the information presented. The legends for panels E and F are also incorrect and It is not clear what "A" and "B" highlighted in panel 2B refer to, which should be clarified. Overall, it is important to ensure that figures are clear, organized, and easy to understand for readers.

 Answer: Thank you very much for the suggestions. We have revised the figures according to your advices.

4.5- It is important to have a Figure showing the chemical structure of chondroitin sulfate from shark skin for better visualization.

 Answer: Thank you very much for the suggestions. We have added the chemical structure of CHS in figure 3.

4.6- The authors can better discuss the pharmacological effect observed with other works in the literature using fucosylated chondroitin sulfate from marine invertebrates or even compare the anti-inflammatory activity of chondroitin sulfate from mammals. Is there any hypothesis about the mechanism of action? The discussion is very poor in this aspect. 1- Liu, H.H.; Ko,W.C.; Hu, M.L. Hypolipidemic Effect of Glycosaminoglycans from the Sea Cucumber Metriatyla Scabra in Rats Fed a Cholesterol-Supplemented Diet. J. Agric. Food Chem. 2002, 50, 3602–3606. 2- Yin, J.; Wang, J.; Li, F.; Yang, Z.; Yang, X.; Sun, W.; Xia, B.; Li, T.; Song, W.; Guo, S. The Fucoidan from the Brown Seaweed: Ascophyllum Nodosum Ameliorates Atherosclerosis in Apolipoprotein E-Deficient Mice. Food Funct. 2019, 10, 5124–5139. 3- Hu, S.; Chang, Y.; He, M.; Wang, J.; Wang, Y.; Xue, C. Fucosylated Chondroitin Sulfate from Sea Cucumber Improves Insulin Sensitivity via Activation of PI3K/PKB Pathway. J. Food Sci. 2014, 79, H1424–H1429. 4- Hu, S.; Chang, Y.;Wang, J.; Xue, C.; Li, Z.;Wang, Y. Fucosylated Chondroitin Sulfate from Sea Cucumber in Combination with Rosiglitazone Improved Glucose Metabolism in the Liver of the Insulin-Resistant Mice. Biosci. Biotechnol. Biochem. 2013, 77, 2263–2268.

Answer: Thank you very much for the suggestions. We have added the discussions in line 212-218.

Thank you for your consideration. I look forward to hearing from you.

Best wishes

Yours sincerely,

Shiwei Hu

Round 2

Reviewer 1 Report

Thank you for your reply.  The paper now is acceptable.

Reviewer 2 Report

The authors responded reasonably to the suggestions.